# Dynamic growth risk of incidentally detected gallbladder polyps–A retrospective, single-center analysis

Sophia Heinrich[1], Piet Janko ten Thoren[1], Patrick Behrendt[1,2,3], Jakob Hagenah[1], Heiner Wedemeyer[1], Andrej Potthoff[1], Benjamin Maasoumy[1]*

**1** Klinik für Gastroenterologie, Hepatologie, Infektiologie und Endokrinologie, Medizinische Hochschule Hannover, Hannover, Deutschland, **2** Institute of Experimental Virology, Twincore, Centre for Experimental and Clinical Infection Research, Hannover, Germany, **3** A Joint Venture between the Medical School Hannover (MHH) and the Helmholtz Centre for Infection Research (HZI), Hannover, Germany

* maasoumy.benjamin@mh-hannover.de

## Abstract

### Background

Size of gallbladder polyps (GP) is considered as a relevant risk factor for neoplastic polyps. However, the definitive impact is an ongoing debate. Current German and European guidelines recommend surveillance for GP > 6 mm and cholecystectomy for GP > 10 mm over a period of two to five years. We aimed to analyze the dynamic growth of gallbladder polyps.

### Methods

Patients at Hanover Medical School who underwent sonography from 2001 to 2020 were retrospectively evaluated for growth rate (GR) of detected GP independent of the underlying primary disease. Only patients with at least one follow-up as well as accurate GP size data were included in the study.

### Results

A number of 253 patients with GP were eligible. Median follow-up was 66 months (24–209 months). Median GR was −0.3 mm/year (IQR 0.79). A subgroup analysis (polyp size 6–10 mm) showed a positive GR in 20.3% of the cases with a median GR of 0.09 mm/year (IQR 0.17). Of note, in 46% of the patients GP were not detectable at follow-up exam. Overall, two patients reached the indication for cholecystectomy (0.8%), whereas only a single patient developed histologically confirmed gallbladder cancer (0.4%). Logistic regression analysis did not reveal any risk factors associated with GP growth.

**Data availability statement:** All relevant data are within the Supporting Information files.

**Funding:** The author(s) received no specific funding for this work.

**Competing interests:** The authors have declared that no competing interests exist.

**Abbreviations:** BMI, body mass index; CHE, cholecystectomy; DGVS, Deutsche Gesellschaft für Gastroenterologie, Verdauungs- und Stoffwechselerkrankungen; EAS, European Association for Endoscopic Surgery; EASL, European Association for the study of the liver; EFISDS, International Society of endoscopic surgery – European Federation; ESGAR, European Society for Gastrointestinal and Abdominal Radiology; ESGE, Europeans Society for Gastrointestinal Endoscopy; GP, gallbladder polyp; GR, growth rate; PSC, primary sclerosing cholangitis.

## Conclusion

The majority of GP, which should be monitored within the current follow-up strategy, are no longer detectable sonographically over time or show a decreasing growth behavior. Only a minority shows a very slow positive GR and only a minority of patients develop malignancy.

## Introduction

Gallbladder polyps (GP) are often detected as incidental finding on abdominal ultrasound examinations. The prevalence has been reported to be between 0.3 to 12.3% [1–7]. Most of them are benign. Of suspected gallbladder polyps 70% are pseudopolyps, which include cholesterol polyps, adenomyomatosis and inflammatory polyps, but most importantly have no malignant potential [8,9]. True GP are mainly adenomas, while malignant polyps are usually adenocarcinomas [10]. In contrast to colorectal cancer, there is no proven adenoma-carcinoma sequence, although there are some studies that suggest such a natural course [11,12]. A recent retrospective study analyzing patients with gallbladder cancer could not confirm an increased risk of cancer in patients with GP [13]. However, several other studies documented presence of GP as a significant risk factor for neoplastic polyps, and the correct differentiation of these two entities is challenging and therefore an ongoing debate [14–17]. As biopsy and histological classification are not feasible, there is a need to define criteria for the decision to perform cholecystectomy (CHE). A size greater than 10 mm is a generally accepted indication for surgery. Data from CHE studies have shown that polyps over 10 mm are neoplastic in 50% of the cases [18–20]. In addition, a relative growth rate over 50% and in some studies also age, although the cut-off varies, Primary Sclerosing Cholangitis (PSC), ethnicity or morphology have been described as risk factors for predicting malignancy [21]. However, the level of evidence is low and the impact of polyp growth remains controversial [2,10,22–28]. A recent meta-analyses confirmed that the data on cancer risk are heterogenous and that, in general, the evidence for a clinical relevance of polyps smaller than 10 mm is rather low [29].

The current Deutsche Gesellschaft für Gastroenterologie, Verdauungs- und Stoffwechselerkrankungen (DGVS) and European Association for the study of the liver (EASL) guideline recommend CHE for GB polyps over 10 mm and surveillance sonography for GP 6–10 mm in size every 3–6 months for at least 5 years [19,22,23,30]. GP under 6 mm are less likely to be associated with malignancy [26,31]. However, interdisciplinary management of GP is controversial. The very precise joint guidelines of the European Society for Gastrointestinal and Abdominal Radiology (ESGAR), the European Association for Endoscopic Surgery (EAS), the International Society of endoscopic surgery – European Federation (EFISDS) and the Europeans Society for Gastrointestinal Endoscopy (ESGE) also recommend an extensive surveillance, including also GP with less than 6 mm for at least 2 years and CHE if the patient is over 60 years of age, has PSC, is of Asian ethnicity or the GP has a sessile morphology [7,10]. If the GP increases in size by more than 2 mm, CHE

is recommended, if the polyp disappears, cessation of surveillance is strongly recommended [10]. However, surveillance interval in case of no polyp growth is only recommended for two years. However, this topic is an ongoing debate. A very recent review from 2025 from the Korean Society of Abdominal Radiology summarizes surveillance recommendations of incidentally detected GP and also gives classification recommendations for those polyps [32].

Some studies consider the growth rate of GP as a risk factor for neoplastic polyps. However, none of these studies has described the incidence on continuous growth of the polyps. A recent study performed in an Asian cohort reported that a yearly growth rate of 3 mm should be considered as a risk factor for neoplastic polyps [2]. However on the contrary, several studies in patients undergoing CHE could not confirm an association between growth rate and malignancy [29,33,34]. However, none of these studies have described the incidence of continuous growth of these polyps.

Gallbladder cancer has a low incidence but a poor prognosis once it is has reached an advanced stage (5 year overall survival in stage II is 28% and 8% or less once it has reached stage III or higher). If detected before infiltration of the muscularis propria, 5 year survival rates can achieve 80% [6]. Given the low incidence but poor prognosis, accurate screening is crucial, but unnecessary diagnostic testing should be avoided, if they do not offer a significant benefit for the patient but cost a lot of resources and might lead to unnecessary CHEs. One review article reports a deficiency of preoperative diagnostic features and summarizes age, tumor markers (CA19–9, CEA, CA-125) as well as GP, porcelain gallbladder and common bile duct dilatation (e.g.,) as potential preoperative risk factors for malignancy [35]. However, we need the right tools to identify suspicious lesions early. We aimed to investigate the common growth rate of incidentally detected gallbladder polyps to assess the need for a strict surveillance of these patients.

## Methods

A total of 253 patients treated at the Hanover Medical School, Germany, between January 2001 and December 2020 were retrospectively included in this study. Patients were managed for different underlying conditions and subsequently received a comprehensive abdominal sonographic evaluation. Standard operation procedure at the Hanover Medical School includes and ultrasound screening of the gallbladder independent of the indication for sonography. All reports of examinations of the abdomen and the liver performed by the in-clinic sonography department were screened for findings of gallbladder polyps. Inclusion criteria were GP at baseline and precise reports of polyp size and at least one follow-up examination. Patients, whose reports did not have a precise polyp size in mm or did not have at least one follow-up in our ultrasound unit, have been excluded from this study. Patients were followed up until last contact and documented size of the polyps was analyzed. Demographic and clinico-pathological data including age, sex, body mass index (BMI), underlying metabolic disorders, liver diseases and malignancies have been assessed for every patient. Growth rate of polyps has been defined as difference between size at baseline and size at end of follow-up divided by the time interval of follow-up periods. Data have been assessed between January 2021 and August 2022.

Statistical analysis was performed using Graphpad Prism Software (Version 8.3.0, Graphpad, USA) and R Studio (version 1.2.5019, R Foundation for Statistical Computing, Austria). For comparison of individual groups median and percentiles were calculated. Predictive ability was assessed by logistic regression analysis (R Studio, glm function). P-values <0.05 were considered statistically significant.

## Ethics

This retrospective study was approved by the local ethic committee of the Hanover Medical School (No. 9133_B0_K_2020) and conducted in compliance with good clinical practice as well as in accordance with the declaration of Helsinki. Patients gave their written consent for their data to be used for research purposes. For data acquisition authors had access to information that could identify individual participants during data collection. After data collection, data have been pseudonymized for analysis.

## Results

### Patient Baseline Characteristics

A total of 253 patients were retrospectively enrolled and analyzed at the ultrasound unit of the Hanover Medical School. Patient cohort is shown in Table 1. Patients were 52.5% female and 47.5% male. The majority of the patients had an underlying chronic liver disease, with fatty liver disease representing the majority of those. Median polyp size was 4.6 mm

**Table 1. Patient cohort.**

|  | patient cohort |
|---|---|
| **total number** | **253** |
| gender (%) |  |
| female | 52,5 |
| male | 47,5 |
| age (years, median (IQR)) | 52 (44; 60) |
| follow up (months, median) | 66 |
| secondary diagnoses (%) |  |
| fatty liver | 63,8 |
| cirrhosis | 14,8 |
| biliary liver disease (PSC/PBC) | 11,3 |
| hepatitis b infection | 14,6 |
| hepatitis c infection | 20,9 |
| risky alcohol consumption | 7,8 |
| autoimmune hepatitis | 7,4 |
| risk factors (%) |  |
| BMI |  |
| <25 | 53,4 |
| >25 | 44 |
| >35 | 2,6 |
| smoking | 19,1 |
| hypertension | 24,1 |
| hyperlipoproteinemia | 6,6 |
| diabetes mellitus | 8,6 |
| size, mm (median (IQR)) | 4.1 (3.1; 5.9) |
| number (median (IQR)) | 1 (1;3) |
| solitary (%) | 58,5 |
| multiple (%) | 41,5 |
| growth rate (mm/year) | −0,3 |
| medium relative growth rate (%) | −26,36 |
| gallbladder cancer (%) | 0,4 |
| indication for cholecystectomy (%) | 5,9 |
| polyp | 26,7 |
| cholecystitis | 13,3 |
| OLT | 26,7 |
| stones | 26,7 |
| other | 6,7 |

*Abbreviations: primary sclerosing cholangitis (PSC), orthotopic liver transplantation (OLT)*

at baseline and median number 2.7 polyps per individual. In total 192 patients had polyps below 5 mm in size, 59 patients showed a polyp size between 6–9 mm and only 5 patients had a polyp size over 10 mm at baseline. During follow up time 15 patients had a CHE. The majority was based on the diagnosis gallbladder polyp, liver transplantation or gallbladder stones. Due to the retrospective design, the indication for CHE when a polyp >10mm was diagnosed for the first time was not stringently adhered to (N = 4). However, to exclude a possible malignancy, that developed after the here observed study time, these patients were followed up outside the study for another 18–48 months, during which no carcinoma diagnosis was confirmed.

Only one patients has been diagnosed with gallbladder cancer.

## Incidence of positive growth rates in gallbladder polyps

Patients were followed up for a median time of 66 months. During this time period a median growth rate of −0.3 mm/year (interquartile range (IQR) 0.79) could be detected, which reflects a median relative growth rate of −8% per year (IQR 18.5) of each patient. Fig 1 demonstrates the individual growth dynamics of each patient (Fig 1A). Even in patients who should be undergoing surveillance according to current guidelines (polyp size 6–10 mm) no significant growth rate was observed (median absolute size difference −2.2 mm (IQR 5.7), median growth rate −35% per year (IQR 1.15), Fig 1B). To exclude that a significant polyp growth is not masked by a large number of disappearing polyps we analyzed as a subgroup all polyps with a size of more than 6 mm and an absolute growth rate higher or equal to 0 mm. This subgroup encompasses 20.3% of the patients within the > 6mm group, demonstrating that almost 80% do not show a positive growth behavior. Median growth rate within this subgroup was calculated as 0.09 mm/year (IQR 0.17). To further exclude disappearing pseudopolyps, we have analyzed a subgroup of patients (N = 205) with at least two independent examinations, that have shown GPs. Again, we could detect a negative median growth rate of −21.5% (IQR 0.75).

## Risk of dynamic growth and malignancy depending on the initial GP size

Since previous data have demonstrated the risk of malignant transformation depends on a total growth rate of over 50%, we analyzed the polyp growth rate of individual patients from baseline to last follow-up [2]. Only N = 16 (6.3%) of the patients showed a growth rate of more than 50%. 11.5% (N = 29) had a moderate increase in size and 4% (N = 10) were completely stable. Most importantly, in the absolute majority of the cases a decrease in size (N = 72, 28.5%) or even complete disappearance of the polyps was observed (N = 116, 45.8%) (Fig 1C).

Looking at the growth of polyps in relation to the initial size of the polyps, we documented that small polyps below 5 mm disappeared or at least decreased in size in 72.1% of the cases (Fig 1D). The same occurred in medium-sized polyps in 79.6% of the cases, which are subject to surveillance according to the current guidelines. However, it is remarkable that also these medium 6−10 mm size polyps showed hardly any growth of more than 50%. In detail, 35% (N = 21) of the polyps have not been detected anymore and 44% (N = 26) showed a decrease in size in the follow-ups (Fig 1D). In the case of large polyps over 10 mm, a significant growth rate could not be detected in any case (with an overall small number of cases in this category, N = 5). Importantly, there is no significant difference in growth rate between polyps below 6 mm and between 6 and 10 mm (p = 0.85). The median annual growth rate for polyps below 6 mm is −0.08 mm/year and for polyps between 6−10 mm is −0.05 mm/year (data no shown).

Most importantly, the incidence of malignancy was 0.4% (N = 1) in this cohort. Of this single malignant polyp, size at first diagnosis was 4.7 mm showing a growth rate of 2.7 mm/year (Fig 1E).

## Predictive factors for dynamic growths

Next, we aimed to identify independent risk factors associated with growth of gallbladder polyps. We performed logistic regression including well-known risk factors such as age, gender, metabolic disorders or underlying liver disease in the

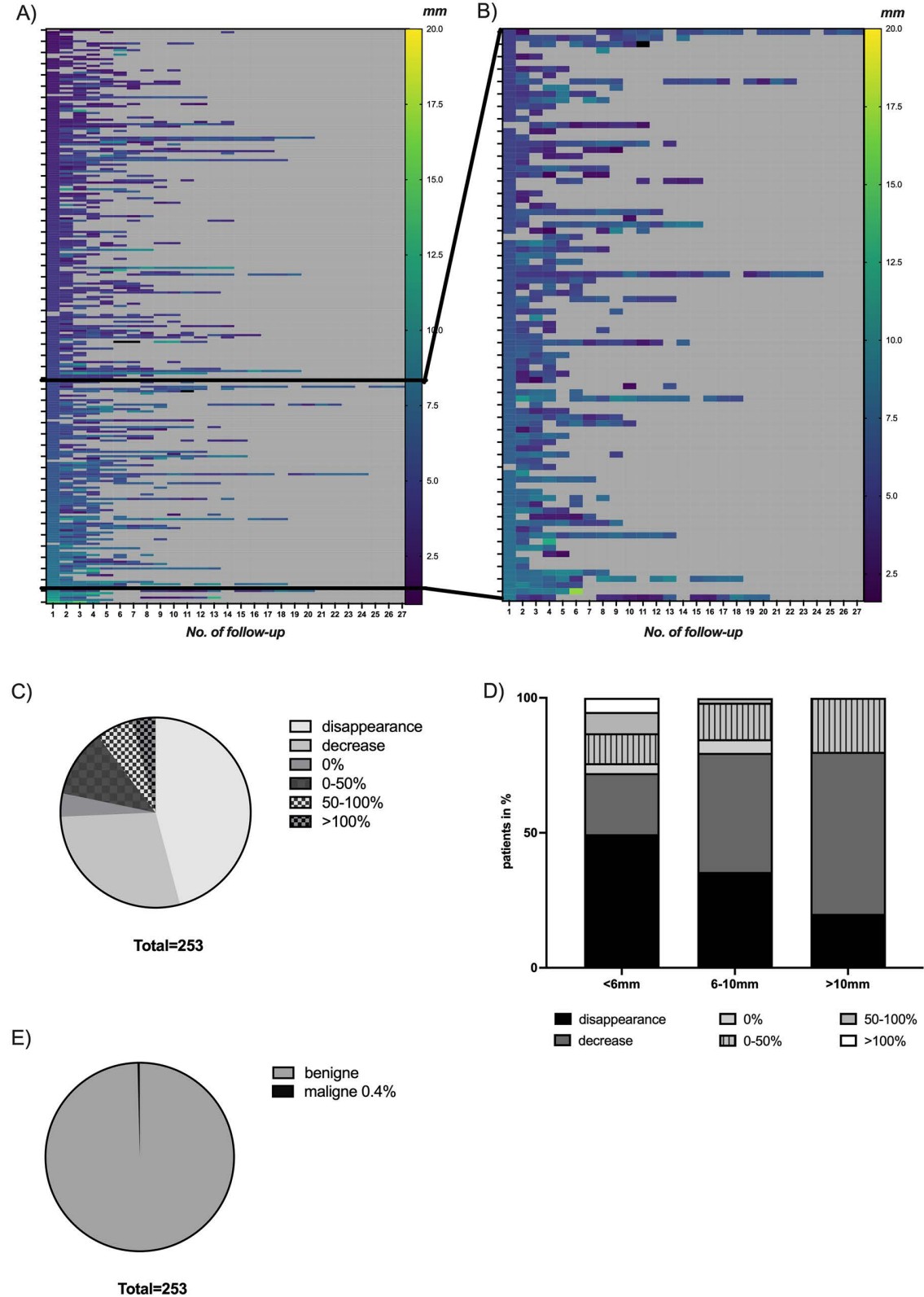

**Fig 1. A) Heatmap demonstrating polyp size in mm of each individual patient (y axis) at every follow-up (x axis).** Each row shows the polyp size of one patient at the respective follow-up (No. on the x axis). The actual size is color coded (from deep purple 0 mm to yellow 20 mm). For each

individual patient the dynamic growth of the polyp over time can be read from left to right. Example: A patient with an initial polyp size below 2.5 mm, that growth up to 12 mm should start on the left with deep purple and continue with blueish coloring till the last follow-up in a greenish one. **B)** Heatmap showing polyp size only of patients with size of polyps being between 6-10 mm at enrollment. **C)** Growth rate of individual polyps in percentage. **D)** Sub-group analysis showing growth rates in percentage depending on size detected at baseline. **E)** Incidence of malignancy.

analysis (Table 2). However, none of the here analyzed parameters showed an independent prognostic impact on growth rate.

## Discussion

Our retrospective analysis included a cohort of 253 patients with gallbladder polyps. We evaluated the growth rate (GR) of all detected polyps, independent of the underlying primary disease, using absolute and relative growth numbers. The median follow-up duration was 66 months, providing valuable longitudinal data for assessing the growth behavior of these polyps. Our results demonstrate a median growth rate of −0.3 mm/year, indicating an overall decrease in polyp size over time. This raises the question if initial GP are actually disappearing pseudopolyps, that do not necessarily require an extensive follow-up.

Our finding aligns with previous studies suggesting that a substantial proportion of gallbladder polyps become undetectable sonographically or exhibit a decreasing growth pattern. It is important to note that a significant number of the included patients had polyps within the size range recommended for surveillance, rather than immediate cholecystectomy. These findings raise questions about the necessity and optimal timing of intervention for such polyps in the context of a cost/benefit consideration.

In our subgroup analysis, which focused on polyps sized between 6 mm and 10 mm, we observed a positive growth rate in 20% of cases, with a median growth rate of 0.09 mm/year. Although this growth rate is relatively slow, it indicates that indeed a minority of polyps may exhibit a long-term growth trajectory, however with a very slow yearly growth rate. It is noteworthy that in the entire cohort, only one patient was diagnosed with gallbladder cancer during the follow-up period. Indeed, indication for cholecystectomy was given by a dynamic polyp growths from 4.7 mm to 10 mm over a 26 months

**Table 2. Logistic regression analysis of potential risk factors associated with GP growth.**

| Parameter | | Univariate analysis | |
|---|---|---|---|
| | Odds ratio | P value* | 95% CI* |
| Age | 0,98 | 0,34 | 0.96; 1.01 |
| Gender | 0,71 | 0,26 | 0.38; 1.29 |
| Number of Polyps | 1,06 | 0,25 | 0.96; 1.16 |
| Fatty liver | 1 | 0,98 | 0.99; NA |
| Liver disease | 1,09 | 0,85 | 0.47;2.86 |
| Risky alcohol consumption | 0,99 | 0,2 | 0.99; 1.00 |
| BMI | 0,99 | 0,91 | 0.94; 1.06 |
| Diabetes mellitus | 0,99 | 0,54 | 0.99; 1.00 |
| Hyperlipoproteinema | 1,11 | 0,86 | 0.30;3.28 |
| Biliary disease (PSC/PBC) | 0,72 | 0,52 | 0.23; 1.84 |
| Gamma-glutamyltransferase | 0,99 | 0,63 | 0.99;1.00 |
| Smoking | 0,99 | 0,36 | 0.99; 1.00 |

*P value and confidence intervals have been calculated using glm function for logistic regeression in R Studio.

*PSC-primary sclerosing cholangitis; PBC – primary biliary cholangitis.*

period. Indication for ultrasound examination of this particular patient was an inflammatory bowel disease. Besides a steatohepatitis the patient had no further underlying liver disease. These findings highlight the need for individualized risk assessment and careful consideration of the potential benefits and risks associated with intervention.

Our logistic regression analysis did not identify any specific risk factors associated with gallbladder polyp growth. However, a recent study showed that presence of GP is increased in NAFLD patients, whereas the authors did not analyze GP growth in this cohort [36]. Further research is needed to identify additional risk factors or biomarkers that can better predict the neoplastic potential of gallbladder polyps and guide decision-making regarding surveillance and cholecystectomy.

The majority of the included patients did not exhibit significant growth over time, leading to the possibility of unnecessary surveillance and potential associated costs and patient anxiety. Therefore, it is crucial to reassess the recommended control intervals and consider individualized approaches based on risk stratification. Other studies have suggested scoring systems, including polyp size, blood flow signal and additional patient characteristics [34,37]. However, polyp growth rate over time has not been assessed in detail. There is only one study investigating prevalence of gallbladder polyps in a follow-up eleven years later. Indeed, they found an increase in polyp number, but not necessarily in size [38]. The overall aim is risk stratification in terms of malignancies. Other approaches than repetitive ultrasounds might be a one-time CT scan or contrast-enhanced ultrasound, maybe extended by a deep learning algorithm, to exclude patients from further follow-ups [16,17,39]. However, there is evidence from a randomized controlled trial that a CT scan might have a lower sensitivity in terms of accuracy for GP than a high-resolution ultrasound examination [40]. There has also been a very recent review that only suggests additional CT or MRI for patients with suspected malidnant GP, with limited sonic window or patients who are already scheduled for CHE [32].

Limitations of this study certainly result from the cohort composition of a tertiary clinical center, the single-center analysis as well as a retrospective data collection. In addition, due to the retrospective survey, ultrasound quality of the different investigators must be mentioned as a possible limitation. Even though polyp size measurement is generally performed at the largest diameter of the polyp and from the inner gallbladder wall till the upper tip of the polyp, measurement can still vary in a certain range between observer as well as due to ultrasound resolution. However, due to the established ultrasound protocol at the Hanover Medical school, the gallbladder has been focused on and if not described, patient have been excluded from this study. Even though polyp size measurement is generally performed at the largest diameter of the polyp and from the inner gallbladder wall till the upper tip of the polyp, measurement can still vary in a certain range between observer as well as due to ultrasound resolution. However, due to the established ultrasound protocol at the Hanover Medical school, the gallbladder has been focused on and if not described, patient have been excluded from this study.

Due to the retrospective design, the indication for CHE when a polyp >10mm was diagnosed for the first time was not stringently adhered to (N=4). However, to exclude a possible malignancy, that developed after the here observed study time, these patients were followed up outside the study for another 18–48 months, during which no carcinoma diagnosis was confirmed. We cannot exclude the development of a potential malignancy after the observed study time. However, the median follow-up time of our patients was 66 months, which exceeds the proposed surveillance of common guidelines. A further limitation of this study is a probably relatively homogenous ethnicity. A further limitation of this study is the probably relatively homogenous ethnicity, which is associated with malignancy risk of gallbladder polyps as shown in a large multiethnic analysis [25]. Even though we have not systematically assessed ethnicity, the majority of the patients will be Caucasian. Last, the majority (over 50%) of the investigated cohort suffered from an underlying chronic liver disease, which might be a confounding factor in the analysis of dynamic polyp growth. However, the here performed logistic regression analysis excluded liver disease as well as fatty liver disease and even biliary disease as an independent risk factor for polyp growth.

In conclusion, our study adds to the growing body of evidence suggesting that the majority of incidentally detected gallbladder polyps, falling within the current follow-up strategy, either regress or exhibit a decreasing growth pattern. In

our investigation only a minority of polyps show a very slow positive growth rate. These findings should be confirmed in prospective, multicentric studies and maybe call for a reconsideration of the current guidelines, particularly in terms of the potential benefit for the respective patients. Part of the discussion should be at least the duration of surveillance if no polyp growth is detected in the first follow up. Further research is warranted to refine risk stratification models and identify the optimal management approach for incidentally detected gallbladder polyps.

## Supporting information

**S1 Data. Supp. data: Primary data used for analyses.** Supp. File primary data.
(XLSX)

## Author contributions

**Conceptualization:** Heiner Wedemeyer, Andrej Potthoff, Benjamin Maasoumy.

**Data curation:** Sophia Heinrich, Piet Janko ten Thoren, Jakob Hagenah, Benjamin Maasoumy.

**Formal analysis:** Sophia Heinrich, Patrick Behrendt.

**Investigation:** Sophia Heinrich.

**Methodology:** Sophia Heinrich, Patrick Behrendt.

**Project administration:** Sophia Heinrich, Andrej Potthoff, Benjamin Maasoumy.

**Resources:** Sophia Heinrich.

**Supervision:** Heiner Wedemeyer, Andrej Potthoff, Benjamin Maasoumy.

**Validation:** Sophia Heinrich.

**Writing – original draft:** Sophia Heinrich.

**Writing – review & editing:** Sophia Heinrich, Piet Janko ten Thoren, Patrick Behrendt, Jakob Hagenah, Heiner Wedemeyer, Andrej Potthoff, Benjamin Maasoumy.

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
