## [Decision Letter · Decision Letter 0]

12 Aug 2025

Dear Dr. Sophia Heinrich,

Thank you for submitting your manuscript to PLOS ONE. After careful consideration, we feel that it has merit but does not fully meet PLOS ONE’s publication criteria as it currently stands. Therefore, we invite you to submit a revised version of the manuscript that addresses the points raised during the review process.

We look forward to receiving your revised manuscript.

Kind regards,

Paolo Aurello

Academic Editor

PLOS ONE

Journal Requirements:

Reviewers' comments:

Reviewer's Responses to Questions

**Comments to the Author**

1. Is the manuscript technically sound, and do the data support the conclusions?

Reviewer #1: Yes

Reviewer #2: Yes

2. Has the statistical analysis been performed appropriately and rigorously?

Reviewer #1: Yes

Reviewer #2: Yes

3. Have the authors made all data underlying the findings in their manuscript fully available?

Reviewer #1: Yes

Reviewer #2: No

4. Is the manuscript presented in an intelligible fashion and written in standard English?

Reviewer #1: Yes

Reviewer #2: Yes

Reviewer #1: Some additional comments: while the majority of patients are likely Caucasian, this should be clearly stated and discussed as a limitation given the known association of ethnicity with GP malignancy risk,please provide more context regarding the indications for the five cholecystectomies, especially the one with confirmed cancer.It would also be helpful to consider including recent systematic reviews or meta-analyses if possible to further corrobrate findings

Reviewer #2: I read with interest your study entitled “Dynamic growth risk of incidentally detected gallbladder polyps – a retrospective, single-center analysis.” This retrospective study includes 253 patients with gallbladder polyps (GP) who were followed using ultrasound over a mean period of 66 months. The authors report a median growth rate of -0.3 mm/year across the cohort, and a positive growth rate in 20% of patients with polyps sized 6–10 mm. Notably, only one patient (0.4%) developed gallbladder cancer during the follow-up period. The authors conclude that the majority of polyps demonstrated a decreasing growth trend, while a minority exhibited very slow positive growth.

The manuscript is well written and clearly structured. The English and scientific language are appropriate. The study addresses a relevant clinical gap where existing literature is scarce and of limited evidence quality, adding novelty to the field. However, given the retrospective design and relatively small patient cohort—especially in the subgroup with GP >10 mm—the findings cannot support definitive clinical recommendations.

Below are several points that may help improve the quality and clarity of the manuscript:

Abstract

The aim of the study should be explicitly stated in the abstract.

Methods

In the sentence: “A total of 253 patients treated at the … were retrospectively included in this study,” please clarify what the patients were being treated for.

The reason for ultrasound follow-ups should be clearly described. If follow-up imaging was performed for conditions such as steatohepatitis, it is possible that the focus was not on gallbladder polyps, potentially leading to underdetection. This potential limitation should be acknowledged.

Results

The sentence: “Only one patient reached was diagnosed with gallbladder cancer” should be revised. The word “reached” is unnecessary and should be removed.

Consider performing a separate analysis comparing growth rates between polyp size categories (e.g., <6 mm vs. 6–10 mm vs. >10 mm, or at least <6 mm vs. >6 mm) to determine whether larger polyps exhibit higher growth rates.

Discussion

The first paragraph largely repeats information from the background and should be removed. Ideally, the discussion should begin by briefly summarizing the major findings of the study.

The second paragraph outlines the aim of the study and would be more appropriate in the Introduction section.

In the sentence: “Indication for ultrasound examination of this particular patients was an inflammatory bowel disease,” the word “patients” should be corrected to “patient.”

The statement: “The current German and European guidelines recommend surveillance for gallbladder polyps larger than 6 mm and cholecystectomy for those exceeding 10 mm. However, our findings raise concerns about the potential benefit and cost-effectiveness of such an approach for all patients,” may be overstated. Since only five patients in the study had polyps >10 mm, and no separate analysis was performed for this subgroup, challenging current guidelines about GP>10 mm is not sufficiently supported by the data.

Please provide the number of patients who underwent cholecystectomy for gallbladder polyps during the follow-up period. What criteria were used to select these patients for surgery? How many malignancies were identified postoperatively? This information would be highly valuable for clinical decision-making and should be included in the Results section.

The sentence: “Due to the retrospective design, the indication for CHE when a polyp >10 mm was diagnosed for the first time was not stringently adhered to (N=4),” refers to important data that should be reported in the Results section rather than introduced in the discussion.

**Do you want your identity to be public for this peer review?** For information about this choice, including consent withdrawal, please see our Privacy Policy

Reviewer #1: **Yes: ** Dr.Million Mohammed Asfaw, General and Endocrine Surgeon Wachemo university,NEMMCS Hospital

Assisstant Professor of Surgery

Reviewer #2: **Yes: ** Sepehr Abbasi dezfouli

---

## [Author Response · Author response to Decision Letter 1]

13 Nov 2025

Reviewer #1:

1. Some additional comments: while the majority of patients are likely Caucasian, this should be clearly stated and discussed as a limitation given the known association of ethnicity with GP malignancy risk.

Response:

We thank the reviewer for this valuable comment. We agree that the lack of ethnic diversity in our cohort represents a limitation, particularly given the known association between ethnicity and the risk of gallbladder polyp malignancy. We have addressed and discussed this aspect in the revised manuscript (Discussion, line 284 ff).

2. Please provide more context regarding the indications for the five cholecystectomies, especially the one with confirmed cancer.

Response:

In total, there have been 15 cholecystectomies in the entire cohort. Patient data are included in the table about the patient cohort. We have further added a short description in the patient baseline characteristic section (line 143ff).

The only case of malignancy originated from the group with an initial polyp size of 4.7 mm. Twenty-two months later, follow-up ultrasound demonstrated an increase in size to 8 mm. A subsequent ultrasound examination performed four months later showed further growth to 10 mm, after which cholecystectomy was carried out. The initial indication for ultrasound was inflammatory bowel disease. Hepatic steatosis was present as a comorbidity, with no evidence of additional underlying liver disease. This case is also discussed in the Discussion section of the revised manuscript (line 236ff).

3. It would also be helpful to consider including recent systematic reviews or meta-analyses if possible, to further corroborate finding.

Response:

We appreciate this excellent suggestion. We have re-screened the literature for the most recent systematic reviews and meta-analyses on this topic and have incorporated the following key references into the revised manuscript: Chang et al., 2025, Kamaya et al., 2022, and Grikyte et al., 2025.

Reviewer #2:

I read with interest your study entitled “Dynamic growth risk of incidentally detected gallbladder polyps – a retrospective, single-center analysis.” This retrospective study includes 253 patients with gallbladder polyps (GP) who were followed using ultrasound over a mean period of 66 months. The authors report a median growth rate of -0.3 mm/year across the cohort, and a positive growth rate in 20% of patients with polyps sized 6–10 mm. Notably, only one patient (0.4%) developed gallbladder cancer during the follow-up period. The authors conclude that the majority of polyps demonstrated a decreasing growth trend, while a minority exhibited very slow positive growth.

The manuscript is well written and clearly structured. The English and scientific language are appropriate. The study addresses a relevant clinical gap where existing literature is scarce and of limited evidence quality, adding novelty to the field. However, given the retrospective design and relatively small patient cohort—especially in the subgroup with GP >10 mm—the findings cannot support definitive clinical recommendations.

Below are several points that may help improve the quality and clarity of the manuscript:

1. Abstract. The aim of the study should be explicitly stated in the abstract.

Response:

We thank the reviewer for this comment. We have revised the abstract to explicitly state the primary aim of the study, i.e., to analyze the dynamic growth of incidentally detected gallbladder polyps during long-term follow-up.

2. Methods. In the sentence: “A total of 253 patients treated at the … were retrospectively included in this study,” please clarify what the patients were being treated for.

The reason for ultrasound follow-ups should be clearly described. If follow-up imaging was performed for conditions such as steatohepatitis, it is possible that the focus was not on gallbladder polyps, potentially leading to under detection. This potential limitation should be

acknowledged.

Response:

We thank the reviewer for this valid point. We have clarified that, at our university hospital, abdominal ultrasound routinely includes an assessment of the gallbladder regardless of the primary indication for sonography. This standard procedure has now been described in the Methods section (line 107ff).

We have also expanded the limitation section to acknowledge that varying attention of investigators to the gallbladder may have influenced the results (line 268ff).

3. Results. The sentence: “Only one patient reached was diagnosed with gallbladder cancer” should be revised. The word “reached” is unnecessary and should be removed.

Response:

We thank the reviewer for the careful reading. The word “reached” has been removed as suggested.

4. Consider performing a separate analysis comparing growth rates between polyp size categories (e.g., <6 mm vs. 6–10 mm vs. >10 mm, or at least <6 mm vs. >6 mm) to determine whether larger polyps exhibit higher growth rates.

Response:

We appreciate this helpful suggestion. We performed the additional analysis, which showed no statistically significant difference in growth rates between small and medium-sized polyps. We have included a description of these results in the Results section (line 195ff). A corresponding figure will be provided for the reviewer in this response letter but was not added to the manuscript.

5. Discussion. The first paragraph largely repeats information from the background and should be removed. Ideally, the discussion should begin by briefly summarizing the major findings of the study.

The second paragraph outlines the aim of the study and would be more appropriate in the Introduction section.

Response:

We agree with the reviewer’s comment and have adapted the structure of the manuscript accordingly. The redundant background content has been removed, and the aim of the study has been shifted to the Introduction section (line 99ff and 212 ff).

6. In the sentence: “Indication for ultrasound examination of this particular patients was an inflammatory bowel disease,” the word “patients” should be corrected to “patient.”

Response:

We thank the reviewer for carefully pointing this out. The error has been corrected.

7. The statement: “The current German and European guidelines recommend surveillance for gallbladder polyps larger than 6 mm and cholecystectomy for those exceeding 10 mm. However, our findings raise concerns about the potential benefit and cost-effectiveness of such an approach for all patients,” may be overstated. Since only five patients in the study had polyps >10 mm, and no separate analysis was performed for this subgroup, challenging current guidelines about GP>10 mm is not sufficiently supported by the data.

Response:

We fully understand the reviewer’s concerns and have deleted this statement from the manuscript to avoid overstating our conclusions.

8. Please provide the number of patients who underwent cholecystectomy for gallbladder polyps during the follow-up period. What criteria were used to select these patients for surgery? How many malignancies were identified postoperatively? This information would be highly valuable for clinical decision-making and should be included in the Results section.

Response:

We thank the reviewer for this important suggestion. This information is presented in the patient cohort table. We have now added a clarifying sentence in the Baseline Characteristics section. The single case of gallbladder cancer is further described in the Discussion, as it represents an isolated finding rather than a major result (line 236 ff).

9. The sentence: “Due to the retrospective design, the indication for CHE when a polyp >10 mm was diagnosed for the first time was not stringently adhered to (N=4),” refers to important data that should be reported in the Results section rather than introduced in the discussion.

Response:

We agree with the reviewer’s suggestion and have moved this information to the Results section (line 144ff).

---

## [Editor Report · Decision Letter 1]

16 Nov 2025

Dynamic growth risk of incidentally detected gallbladder polyps – a retrospective, single-center analysis

PONE-D-25-14169R1

Dear Dr. Sophia Heinrich

We’re pleased to inform you that your manuscript has been judged scientifically suitable for publication and will be formally accepted for publication once it meets all outstanding technical requirements.

Kind regards,

Paolo Aurello

Academic Editor

PLOS ONE
---

## [Editor Report · Acceptance letter]

PONE-D-25-14169R1

PLOS One

Dear Dr. Heinrich,

I'm pleased to inform you that your manuscript has been deemed suitable for publication in PLOS One. Congratulations! Your manuscript is now being handed over to our production team.

Kind regards,

on behalf of

Dr. Paolo Aurello

Academic Editor

PLOS One